# Deep Learning Prediction of *TERT* Promoter Mutation Status in Thyroid Cancer Using Histologic Images

**DOI:** 10.3390/medicina59030536

**Published:** 2023-03-09

**Authors:** Jinhee Kim, Seokhwan Ko, Moonsik Kim, Nora Jee-Young Park, Hyungsoo Han, Junghwan Cho, Ji Young Park

**Affiliations:** 1Department of Pathology, Kyungpook National University School of Medicine, Kyungpook National University Chilgok Hospital, Daegu 41404, Republic of Korea; 2Clinical Omics Institute, Kyungpook National University, Daegu 41405, Republic of Korea; 3Department of Biomedical Science, School of Medicine, Kyungpook National University, Daegu 41944, Republic of Korea; 4Department of Physiology, School of Medicine, Kyungpook National University, Daegu 41944, Republic of Korea

**Keywords:** *TERT*, thyroid cancer, deep learning, color transformation, CNN, CRNN

## Abstract

*Background and* objectives: Telomerase reverse transcriptase (*TERT*) promoter mutation, found in a subset of patients with thyroid cancer, is strongly associated with aggressive biologic behavior. Predicting *TERT* promoter mutation is thus necessary for the prognostic stratification of thyroid cancer patients. *Materials and Methods:* In this study, we evaluate *TERT* promoter mutation status in thyroid cancer through the deep learning approach using histologic images. Our analysis included 13 consecutive surgically resected thyroid cancers with *TERT* promoter mutations (either C228T or C250T) and 12 randomly selected surgically resected thyroid cancers with a wild-type *TERT* promoter. Our deep learning model was created using a two-step cascade approach. First, tumor areas were identified using convolutional neural networks (CNNs), and then *TERT* promoter mutations within tumor areas were predicted using the CNN–recurrent neural network (CRNN) model. *Results*: Using the hue–saturation–value (HSV)-strong color transformation scheme, the overall experiment results show 99.9% sensitivity and 60% specificity (improvements of approximately 25% and 37%, respectively, compared to image normalization as a baseline model) in predicting *TERT* mutations. *Conclusions*: Highly sensitive screening for *TERT* promoter mutations is possible using histologic image analysis based on deep learning. This approach will help improve the classification of thyroid cancer patients according to the biologic behavior of tumors.

## 1. Introduction

Thyroid cancer is one of the most common malignancies in humans [1]. Although the majority of thyroid cancers show indolent behavior [2], tumor recurrence and distant metastasis can occur [3,4]. The telomerase reverse transcriptase (*TERT*) gene, located on chromosome 5p15.33, is involved in telomere maintenance and associated with cellular senescence [5]. *TERT* promoter mutations have been repeatedly found in human cancer, particularly with high frequency in human melanoma and thyroid cancer [5,6]. Furthermore, *TERT* promoter mutations C228T and C250T have been known to occur quite frequently (mutation hotspots) [7,8]. Notably, *TERT* promoter mutations in thyroid cancer have been associated with aggressive clinical behavior [9,10,11]. Thus, the detection of *TERT* promoter mutations is important for prognostic stratification and patient management.

Evidence has shown that digital pathology with artificial intelligence (AI) can have a wide range of applications [12]. In fact, the use of digital pathologic images can improve quantitative analysis of certain histologic features, such as tumor-infiltrating lymphocytes [13]. Furthermore, current studies have been actively investigating methods for predicting the mutation status of genes with diagnostic and therapeutic implications using digital pathologic images [14,15,16,17]. Conventionally, a two-step approach is used for predicting genetic alternations in various cancer types [14,15,18]. First, typical tumor areas are distinguished using tissue slides, and subsequently, another deep neural network is applied to classify mutations at the tile level within tumor areas. Recent advances include attention-based multiple-instance learning performed by aggregating tile features with weight-scoring values learned by a neural network for slide-level prediction [19,20,21].

It is important to consider the color diversity of histopathological images when training AI for better tumor classification. Recent studies have introduced deep neural networks along with color normalization or transformation methods [22,23] as an image preprocessing step that reduces the generalization error.

To the best of our knowledge, this study is the first to evaluate the mutation status of the *TERT* promoter in thyroid cancer using our deep learning model. On the basis of the general perspective of medical doctors, we designed a two-step cascaded architecture to predict the mutation status of the *TERT* promoter in thyroid cancer. In the first step, the architecture predicted tumor areas using color transformation methods and convolutional neural networks (CNNs). To subsequently infer the *TERT* promoter mutation status, the combination of a CNN and recurrent neural network (RNN) model (CRNN) [24,25] was applied in the second step, which focuses on finding cell abnormalities associated with *TERT* promoter mutation status.

## 2. Materials and Methods

### 2.1. Study Population

We retrospectively evaluated 80 consecutive surgically resected thyroid cancer cases from 2016 to 2021 whose samples underwent *TERT* promoter polymerase chain reaction (PCR) testing and found 13 (16.3%) cases with *TERT* promoter mutations (either C228T or C250T). *TERT* promoter mutation status was confirmed via real-time PCR at the Department of Pathology, Kyungpook National University Chilgok Hospital. *TERT* promoter PCR testing was mainly performed for older patients (>55 years) with large tumors having widely infiltrative growth patterns and thyroid cancers showing aggressive clinical behavior [6,26]. We then randomly selected 12 surgically resected thyroid cancer cases having a wild-type *TERT* promoter during the same period. Considering the class-imbalance problem in training deep learning [27], we finally selected a number of *TERT*-negative cases that is similar to that of positive ones. The clinicopathologic data of the patients were retrieved from their medical records. This study was conducted in accordance with the guidelines of the Declaration of Helsinki. The requirement for written informed consent from the patients was waived because of the retrospective nature of the study.

### 2.2. Histologic Evaluation

Surgical specimens were fixed in 10% neutral-buffered formalin and embedded in paraffin blocks. The paraffin blocks were then cut into 4 μm thick sections and stained with hematoxylin and eosin. Two independent pathologists specializing in thyroid pathology (MSK and JYP) reviewed all available slides, and the representative slides were selected for scanning (Figure 1). Tumors were diagnosed and classified according to the fifth edition of the World Health Organization classification of thyroid neoplasms [28].

### 2.3. Dataset Preparation

#### 2.3.1. Annotation of Tumor and *TERT* Positives

Each slide has been annotated according to three types of regions, as shown in Figure 2: normal regions (red contours), tumor regions (yellow contours), and *TERT* regions of interest (ROIs) within the tumor (bounding boxes). The tumor regions have been accurately delineated, while normal regions and *TERT* ROI boxes have been marked partially. Figure 2 shows an example of overall annotation tasks on whole-slide image (WSI) data.

#### 2.3.2. Downsampling Ratio

To analyze patches in different WSI scales, the downsampling level was defined on the basis of the generic equation of 1/$2^{level\;value}$. Thus, the original size was extracted from the level value 0. Because each step focuses on different features, we applied different level values to both steps of the deep learning model. Patch images scaled to ¼ the size of the original image resolution were used to classify tumor areas; however, patches used for predicting *TERT* promoter mutations were extracted using the original resolution.

### 2.4. Whole Architecture for TERT Prediction

We constructed a cascade deep learning model consisting of a tumor classifier and a *TERT* predictor that inferred mutation status according to the tile-based WSI input. Figure 3a shows how the CNN model recognizes tumor areas at low magnification levels with tiling patches. The predicted patches are rescaled to the original level to examine cytologic atypia at high resolutions and are delivered to the CRNN model for the prediction of *TERT* mutation (negative or positive) as shown in Figure 3b.

### 2.5. Data Split

We used 25 WSIs (13 for *TERT* positive, 12 for *TERT* negative) with the given dataset being split into 5 cross-validation sets at the slide level. To evenly split the *TERT*-positive and *TERT*-negative cases in each fold, each positive and negative slide was first separated, after which five-fold cross-validation was performed. Table 1a shows the number of patches for tumor classification. Because each WSI contains various tumor area distributions, each set has a different amount of data. Table 1b shows the distribution of *TERT* ROI bounding boxes made according to *TERT*-negative and *TERT*-positive slides in the second step.

### 2.6. Classification of Tumor Areas Using CNN

#### 2.6.1. Patch Filtering

Given the enormous size of WSI data, each WSI was tiled using a patch size of 256 × 256 at level value 2 (i.e., downsized to ¼ of the original). Some patches include unnecessary components, such as void backgrounds observed as being white. To prevent the use of white background patches, we filtered out void patches on the basis of the grayscale pixel criteria. Each patch image was converted to 8-bit grayscale, and a binary image was generated by setting the valid pixel value threshold to <230 in order to identify the background areas in the patch image. After each patch was inspected using the value at the pixel level, only patches that had a background pixel rate exceeding 40% were excluded from the training dataset.

#### 2.6.2. Color Transformation as Image Preprocessing

To account for the color diversity of the pathological images, such as those acquired from different scanners or using various staining conditions, color transformations, including hue–saturation–value (HSV) and hematoxylin–eosin–DAB (HED) methods, were applied for a better classification performance [22]. Color-augmentation strategies using HSV/HED-light and HSV/HED-strong have been investigated for tumor classification in Figure 4
α=Random Choice[uniform distribution(1−θ, 1+θ)]
β=Random Choice[uniform distribution(−θ, θ)]
$$image^{\prime }=\alpha *image+\beta$$
where α is the slope, β is the intercept parameter, and $image^{\prime }$ is the color-transformed image. For HED-light and HED-strong, θ parameters of 0.05 and 0.2 were applied, respectively. Moreover, a hue value of 0.1 and saturation value of 1.0 were applied for HSV-light and HSV-strong, respectively. The θ, hue, and saturation parameters manipulate how much to jitter the HED or HSV color space. Color normalization using the mean and standard deviation values for the whole-image data was performed in order to apply pre-trained weights from the ImageNet dataset and determine optimal preprocessing methods.

#### 2.6.3. CNN Model Training

For tumor classification, we applied three state-of-the-art CNN models: DenseNet161 [29], VGG16 [30], and EfficientNet_b4 [31]. Figure 5 shows an overview of the CNN training model architecture for tumor area prediction. Each CNN model was implemented using a Pytorch deep learning framework and used a pre-trained model generated on the ImageNet dataset. To address the class-imbalance problem, class weighting, which is the ratio of the number of samples in each class to the total training samples, was applied to the cross-entropy loss function. A total of 3 CNN training models were created on NVIDIA RTX A6000 GPUs, with data being loaded at a batch size of 64.

Because most training performances were saturated or overfitted after the 30th epoch, tasks were forcibly stopped early at that point. In the experiments, the initial learning rate and weight decay were determined to be 5.0e-5 and 1.0e-4, respectively, using an ADAM optimizer to perform a parameter sweep that would derive the best-performing architecture. The best model was screened on the basis of validation accuracy.

Each validation set was evaluated on the best-performing models having the highest accuracy scores. As the trained models were generated using 5 different color transformation methods (i.e., normal, HED-light, HED-strong, HSV-light, and HSV-strong) at each cross-validation set, the experiment had a total of 25 training operations for each CNN model.

### 2.7. Prediction of TERT Promoter Mutation Status Using CRNN

After classifying tumor areas at the first step, the second step predicted whether the patches were positive or negative for *TERT* mutations. Considering the diagnosis of the annotated ROI bounding-box region at high resolution, the bounding box was magnified to the original scale to determine cytologic atypia levels, after which the boxes were cropped into 24 fragments (see Figure 6). Because each annotated box differs in size, the patches were overlapped, with the overlap size being set to automatically fit the corresponding sizes.

To integrate features from the 24 fragments, a CRNN, which is a combination of CNN and RNN, was constructed, after which a multilayer perceptron (MLP) module was created as shown in Figure 6. To extract the features of each patch, we applied ResNet152 as a CNN module and added Long Short-Term Memory (LSTM) (which has three RNN layers) to integrate the features and establish a two-layer MLP module to make the final predictions regarding the *TERT* mutation status. Figure 6 shows that all patches were passed through the CNN module and delivered to the RNN module.

Model training was performed using the CRNN network using NVIDIA RTX A6000 GPUs, with the data being loaded at a batch size of 64. Each CNN module used a pre-trained model generated on the ImageNet dataset. To measure the performance of the model, cross-entropy loss was applied, which yielded the summed outputs. According to training performance tracking, most training performances were saturated or overfitted after the 50th epoch; thus, tasks forcibly stopped early at that point. By sweeping the hyperparameters to derive the best-performing architecture, the learning rate was determined to be 1.0e−3 using an ADAM optimizer, and the best model was selected according to validation accuracy.

## 3. Results

### 3.1. Patient Cohort

The median age of the patients was 53 years (range 22–79 years) with the cohort including 9 males and 16 females. The 13 thyroid cancer cases with a mutant *TERT* promoter comprised 5 conventional papillary thyroid carcinomas, 3 follicular thyroid carcinomas, 1 poorly differentiated thyroid carcinoma, and 3 anaplastic thyroid carcinomas. The 12 thyroid cancer cases with a wild-type *TERT* promoter comprised 8 conventional papillary thyroid carcinomas, 2 follicular thyroid carcinomas, and 2 poorly differentiated thyroid carcinomas. Detailed information regarding the patient cohort is included in Appendix A.

### 3.2. Tumor Classification

Five metrics, namely, precision, recall, f1 score, accuracy, and area under the curve (AUC) score were utilized to evaluate tumor classification performance. Some metrics, such as precision, recall, and f1 score, have two classification results; therefore, their results were macro-averaged over normal and tumor classes.

Figure 7 summarizes the performance results of the tumor classification performed using the DenseNet161, VGG16, and EfficientNet_b4 CNN architecture with image channel-wise normalization (i.e., subtracting the mean and dividing by the standard deviation from ImageNet datasets) and four different color-transformation methods. As five cross-validations were conducted, each result was averaged and its standard deviations indicated as shown in Figure 7.

Most bar plots show that the performance scores of the CNN architecture using the color transformation methods were better than those using image normalization, resulting in an improvement of approximately >6% (±2%) in terms of both accuracy and AUC score. The figures show that DenseNet161, VGG16, and EfficientNet_b4 had the best performance results with HSV-strong, HED-strong, and HSV-light transformation methods, respectively. More detailed results are provided in Appendix A.

### 3.3. TERT Classification Performance Results Using the CRNN Model

As the color transformation methods showed improved results in the first step, we implemented additional experiments using the HSV-strong method in the next step. Table 2 shows the prediction results of *TERT* mutation status as negative or positive using the CRNN (ResNet152 + LSTM) model. Accordingly, an accuracy of 0.92 and AUC score of 0.90 were obtained without applying any color transforms. However, after applying the HSV-strong method, an accuracy of 0.95 and AUC score of 0.94 were obtained, which were noticeably better scores. As shown in Table 2 a,b, all performance metric scores were better when using the HSV-strong method than when using a plain image-normalization scheme.

To examine the areas highlighted by the CRNN model in the tumor patches predicted to be *TERT* positive, we created attention maps for the CNN modules on the basis of the score values extracted from the last fully connected layers. Because the *TERT* ROI has 24 patch images, each attention map is displayed in Figure 8, where the deeper the green color, the higher the attention score. In general, tumor cells showing size enlargement and nuclear hyperchromasia with prominent nucleoli, which are usually associated with aggressive biologic behavior, are concentrated in deep green areas in each attention map.

### 3.4. Whole Inference Process

In this experiment, our cascaded architecture, comprising trained CNN and CRNN, recognizes tumors and finally predicts *TERT* mutation status according to the tile-based WSI input. The model identifies tumor areas at a downsampling level value 2 (¼ downscale) with a patch size of 256 × 256 and then adjusts the predicted patches to level value 0 (original scale) with a size of 1024 × 1024. The higher-resolution patch was cropped to 24 patches with a size of 256 × 256, which were delivered simultaneously to the CRNN model to predict *TERT* mutation status. As shown in Figure 9, the CNN model determined whether each patch belonged to non-tumor or tumor areas. Once classified as a tumor, patches correctly predicted as *TERT* positive were marked with a green color, but those predicted incorrectly were indicated with a purple color.

Table 3 presents the validation results of the whole process using our cascade architecture. Each validation slide passed through the first CNN models (i.e., DenseNet161, VGG16, and EfficientNet_b4) and was retrieved along with the best color transformation methods, such as HSV-strong, HED-strong, and HSV-light. Given that we performed five-fold cross-validation, each result is the mean value of five validation sets. Two different transformation methods, namely, image normalization and HSV-strong, were applied to the CRNN model, and the results of each combination in terms of both sensitivity and specificity are shown in Table 3. Each combination of the CNN and CRNN models along with the HSV-strong transformation provided better performance. Notably, we observed a 23%, 15%, and 6% improvement in sensitivity and a 37%, 22%, and 8% improvement in specificity, respectively, compared to image normalization (Norm) as a baseline model.

## 4. Discussion

To the best of our knowledge, this study is the first to demonstrate that *TERT* promoter mutation status in thyroid cancer is associated with histologic features detectable using the deep learning approach. Through our cascaded deep learning approach, we learned that *TERT* promoter mutation status is associated with tumor cell size enlargement and nuclear atypia with prominent nuclear atypia, which is often associated with aggressive tumor behavior. This is consistent with the results of previous studies, which have shown that *TERT* promoter mutations usually accompany morphological changes [6,26].

Several previous studies have demonstrated the prognostic significance of *TERT* promoter mutations in thyroid cancer. In conventional papillary thyroid carcinoma, *TERT* promoter mutations are often associated with subtypes showing aggressive clinical behavior, including the tall cell [32] and hobnail subtypes [33]. Moreover, differentiated high-grade follicular cell-derived, poorly differentiated, and anaplastic thyroid carcinomas frequently harbor *TERT* promoter mutations [28]. Real-time PCR testing [34] and next-generation sequencing [35] are currently being used to confirm *TERT* promoter mutation status. However, testing all thyroid cancers for the *TERT* promoter mutation might not be cost effective considering the low incidence of *TERT* promoter mutations in thyroid cancer [7]. Therefore, predicting *TERT* promoter mutations via histologic images using a deep learning approach can be a useful screening tool.

Generally, WSI data have different color tones, thus color normalization or transformation has been regarded as an essential step in histopathology image processing. Regarding color normalization, prediction performance can also be influenced by a particular reference template. Manual selection of the templates also adds to the work. Furthermore, a recent study showed that a proper color transformation scheme outperformed the color normalization method [19]. Therefore, the current study focuses on color diversity rather than normalization, which enables us to leverage color transformation to improve *TERT* mutation prediction.

This study has some limitations. First, aberrant functionality of the *TERT* gene can be acquired through other mechanisms, including *TERT* mRNA overexpression and aberrant promoter methylation patterns [36,37,38]. Hence, thyroid cancers designated as having wild-type promoters in the present study might have exhibited abnormal *TERT* functioning, although the possibility is quite small. Pathogenic mutations in the *TP53*, *BRAF*, *RAS*, and other genes can also promote histologic changes associated with thyroid cancer [39,40]. Indeed, among the 13 thyroid cancer cases with *TERT* promoter mutations, 5 cases subsequently underwent targeted next-generation sequencing at the request of the clinician. Other than *TERT* promoter hotspot mutations, pathogenic mutations in *NRAS*, *TP53*, *BRAF*, *RB1* deletion, and *NCOA4*-*RET* fusion were detected in these cases. However, only *TERT* promoter mutations were recurrently detected in these cases. We, therefore, suggest that *TERT* promoter mutations were most closely associated with the histologic findings observed in the present study.

We mainly performed PCR testing for *TERT* promoter mutations in thyroid cancer cases presenting with large tumor sizes showing aggressive behavior [4,26], which might lead to selection bias. Subsequent studies that include a larger number of thyroid cancer cases with a wider morphological spectrum should be performed to further validate the findings presented in this study.

We did not perform a subgroup analysis of thyroid cancer because of the limited number of cases. In a future study, we are planning to consider subgroup analysis to better predict *TERT* promoter mutations according to subtypes, with a larger number of cases.

Intratumoral heterogeneity is a well-known phenomenon in thyroid cancer [41], and *TERT* promoter mutation status can differ across distinct tumor areas. However, *TERT* promoter mutations in four of the five cases who underwent targeted next-generation sequencing were found to be clonal events after considering tumor cellularity and variant allele frequency of the *TERT* promoter mutation (data not included), with only one case having a *TERT* promoter mutation determined to be a subclonal event.

Moreover, a relatively small number of *TERT* promoter mutations and wild-type cancers were used in the current study. Training a deep learning model requires a much larger number of cases. However, *TERT* promoter mutations occur infrequently in thyroid cancer. Thus, collecting a large number of thyroid cancer cases with *TERT* promoter mutations was difficult.

Given our knowledge of the issues regarding case numbers, we focused on smart-sized cases confirmed with PCR testing and an efficient learning approach using transfer learning. Furthermore, this work focuses on tile-level rather than case-level prediction using deep learning. Thus, the number of tile images (Table 1) was sufficient to train the deep learning model. Although many relevant studies assign the same label to every patch in the tumor region of a WSI [14,15,16,17,18], this approach suffers from noisy training [20].

Therefore, to obtain good quality data from a limited number of *TERT*-positive cases, the two pathologists who are experienced in thyroid pathology that were involved in our study conducted fine-grained *TERT* ROI annotation in tumor areas. Using a smart-sized and good data set, the deep learning approach was able to differentiate the morphologic features at the tile level with 0.99 sensitivity for *TERT* mutation positivity.

## 5. Conclusions

High-sensitivity screening for *TERT* promoter mutation status in thyroid cancer is possible through histologic analysis with the assistance of deep learning along with color transformation schemes. Thyroid cancer patients with a high probability of harboring *TERT* promoter mutations can thus be screened for confirmative *TERT* promoter mutation testing, such as real-time PCR or next-generation sequencing, which can ultimately reduce the medical costs shouldered by them. Further studies with a larger cohort might be required to validate the results presented in the current study.

## Figures and Tables

**Figure 1 medicina-59-00536-f001:**
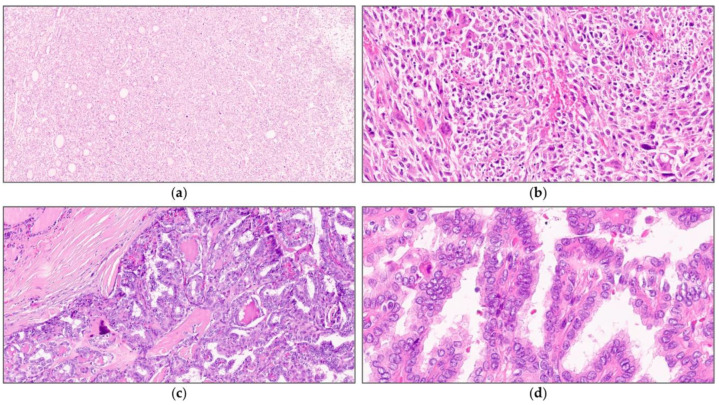
Representative images of thyroid cancers harboring a *TERT* promoter mutation (**a**,**b**) and a wild-type *TERT* promoter (**c**,**d**). Thyroid cancer with a *TERT* promoter mutation shows (**a**) a solid architecture with (**b**) prominent nuclear atypia and frequent mitosis, whereas that with a wild-type *TERT* promoter shows (**c**) conventional papillary architecture with (**d**) a lesser degree of nuclear atypia.

**Figure 2 medicina-59-00536-f002:**
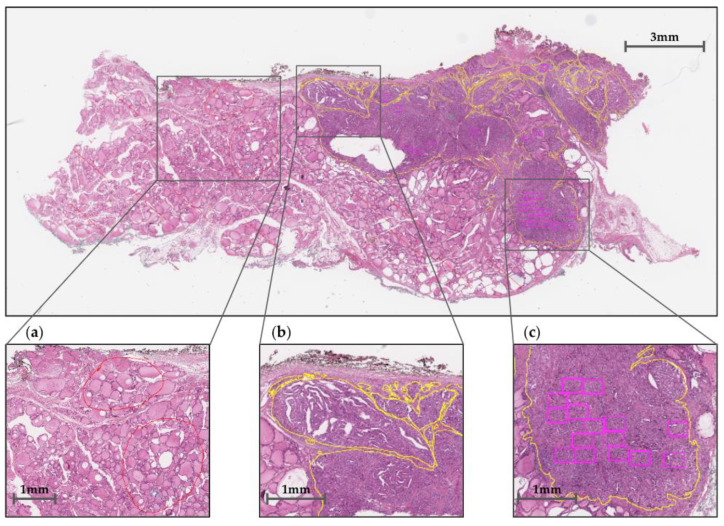
Whole-slide image (WSI) annotation: (**a**) normal regions, (**b**) tumor regions, and (**c**) *TERT* ROIs are marked in red and yellow contours and purple bounding boxes, respectively.

**Figure 3 medicina-59-00536-f003:**
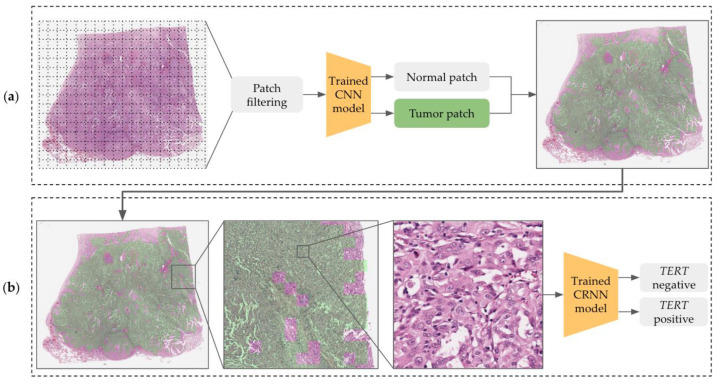
Overview of the *TERT* prediction architecture consisting of (**a**) a tumor classifier and (**b**) a mutation predictor.

**Figure 4 medicina-59-00536-f004:**
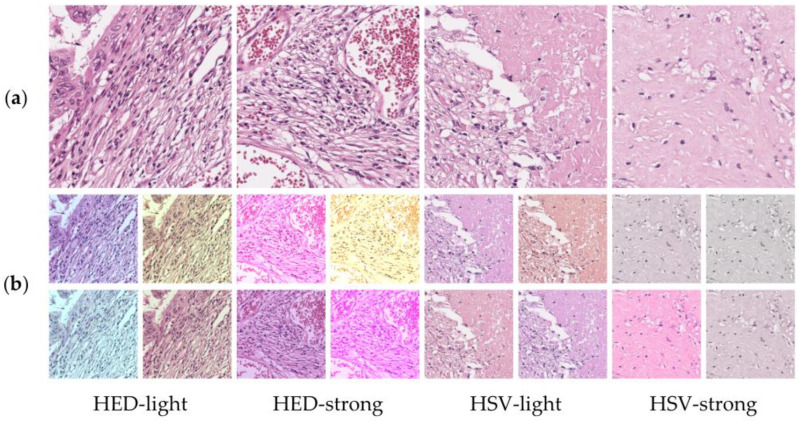
Examples of HED and HSV color transformations: (**a**) original patches and (**b**) color-transformed patches.

**Figure 5 medicina-59-00536-f005:**
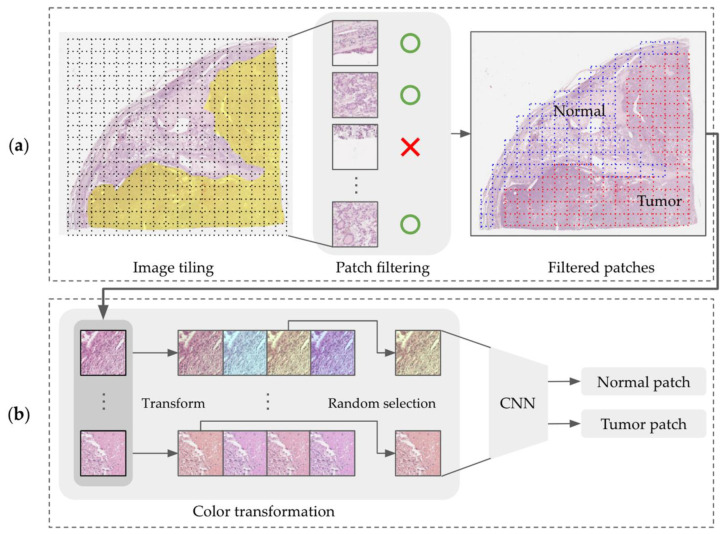
Overview of the CNN training model architecture for tumor area prediction showing (**a**) image tiling of WSI and patch filtering out from the tiled dataset (the yellow mask represents tumor area annotation, blue dot boxes are normal patches, and red dot boxes are tumor patches) and (**b**) the color transformation of the filtered patches.

**Figure 6 medicina-59-00536-f006:**
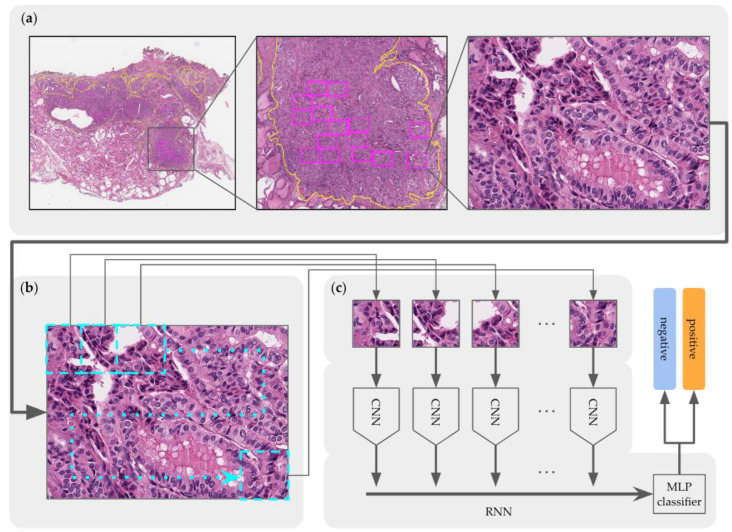
Overview of the CRNN training model architecture for *TERT* mutation prediction containing modules for: (**a**) *TERT* ROI bounding-box extraction, (**b**) tiling to 256 × 256 patches, and (**c**) CRNN.

**Figure 7 medicina-59-00536-f007:**
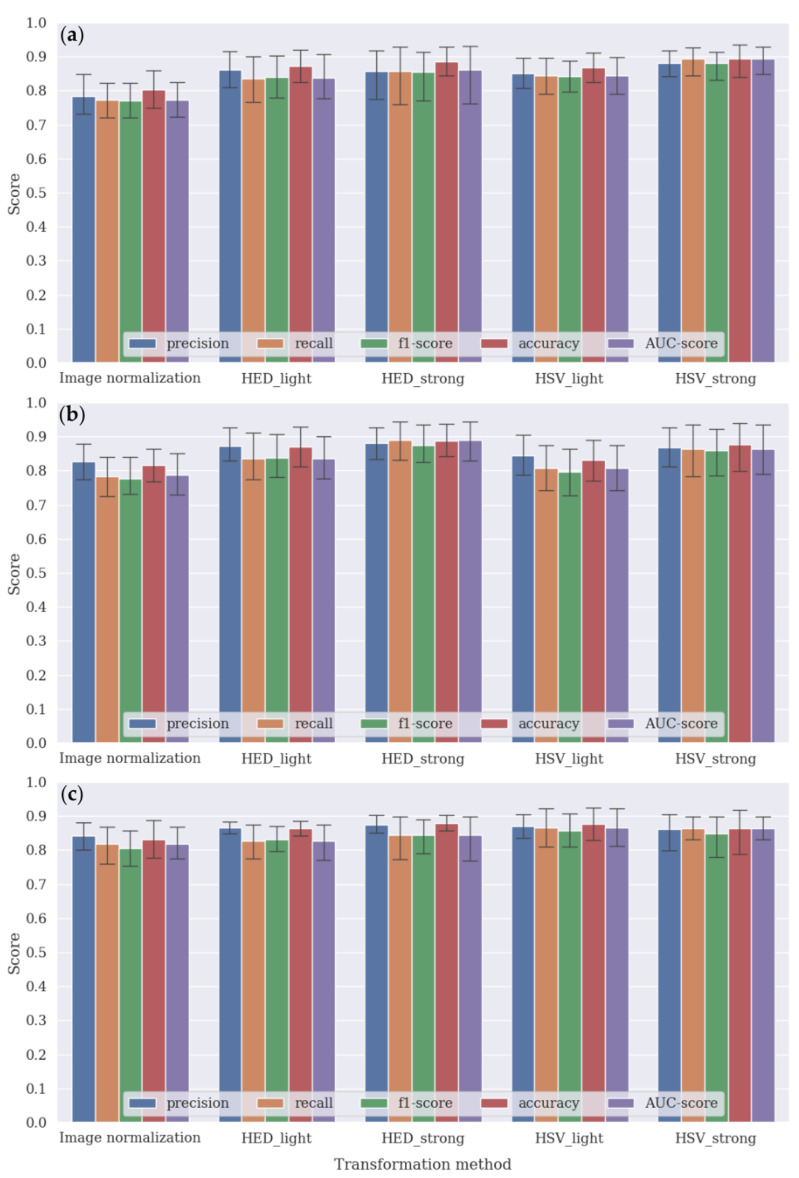
Tumor classification results obtained via five different transformation methods using the (**a**) DenseNet161, (**b**) VGG16, and (**c**) EfficientNet_b4 CNN models.

**Figure 8 medicina-59-00536-f008:**
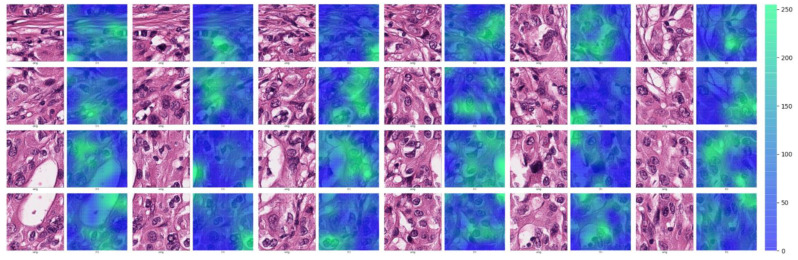
Attention maps of *TERT*-positive cases on the CRNN model.

**Figure 9 medicina-59-00536-f009:**
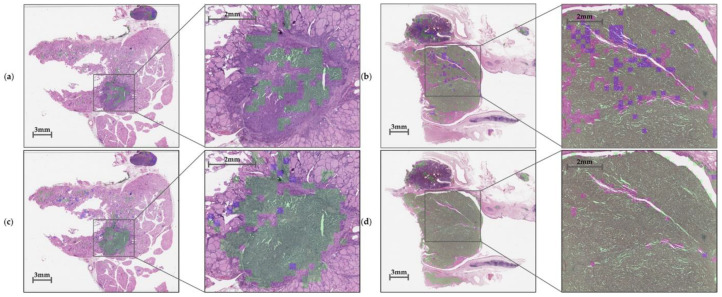
Inference results for *TERT* based on (**a**) negative and (**b**) positive cases without color transformations (as a base model) and (**c**) negative and (**d**) positive cases with color transformations. True positive *TERT* predictions are marked with a green color, whereas all others are indicated with a purple color.

**Table 1 medicina-59-00536-t001:** Training–validation data split information for tumor classification and *TERT* prediction: (a) data counts for tumor and non-tumor patches and (b) *TERT*-negative and *TERT*-positive ROI boxes in each cross-validation (CV) set.

	(a) Tumor and Non-Tumor Patches				(b) *TERT* ROI for Negative and Positive Cases			
	Training		Validation		Training		Validation	
	Normal	Tumor	Normal	Tumor	Negative	Positive	Negative	Positive
CV Set 1	26794	26417	10099	5699	145	225	45	83
CV Set 2	24963	27675	11930	4441	149	239	41	69
CV Set 3	28618	24019	8275	8097	143	236	47	72
CV Set 4	34142	20825	2751	11291	168	246	22	62
CV Set 5	33055	29528	3838	2588	155	286	35	22

**Table 2 medicina-59-00536-t002:** Results of *TERT* mutation prediction on the CRNN (ResNet152 + LSTM) model showing the mean and standard deviation values following five-fold cross-validation.

	(a) CRNN with Normal Transform			(b) CRNN with HSV-Strong		
	Precision	Recall	f1-Score	Precision	Recall	f1-Score
Negative	0.93 (±0.13)	0.84 (±0.19)	0.87 (±0.12)	0.97 (±0.03)	0.89 (±0.18)	0.92 (±0.11)
Positive	0.93 (±0.09)	0.96 (±0.08)	0.94 (±0.07)	0.95 (±0.09)	0.98 (±0.02)	0.96 (±0.04)
Accuracy			0.92 (±0.08)			0.95 (±0.06)
AUC score			0.90 (±0.09)			0.94 (±0.08)

**Table 3 medicina-59-00536-t003:** Inference results for the cascaded CNN + CRNN(ResNet152+LSTM) architectures.

Methods	Sensitivity	Specificity
DenseNet161(Norm) + CRNN(Norm)	0.76 (±0.43)	0.23 (±0.18)
DenseNet161(HSV-strong) + CRNN(Norm)	0.96 (±0.12)	0.55 (±0.32)
**DenseNet161(HSV-strong) + CRNN(HSV-strong)**	**0.99 (±0.00)**	**0.60 (±0.31)**
VGG16(Norm) + CRNN(Norm)	0.78 (±0.34)	0.33 (±0.29)
VGG16(HED-strong) + CRNN(Norm)	0.89 (±0.28)	0.37 (±0.31)
**VGG16(HED-strong) + CRNN(HSV-strong)**	**0.93 (±0.26)**	**0.50 (±0.31)**
EfficientNet_b4(Norm) + CRNN(Norm)	0.92 (±0.22)	0.51 (±0.30)
EfficientNet_b4(HSV-light) + CRNN(Norm)	0.95 (±0.12)	0.50 (±0.34)
**EfficientNet_b4(HSV-light) + CRNN(HSV-strong)**	**0.98 (±0.05)**	**0.59 (±0.26)**

## Data Availability

All data generated and analyzed during this study are included in this article and its Appendix A.

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
