# Peer review of "Deep Learning Prediction of TERT Promoter Mutation Status in Thyroid Cancer Using Histologic Images"

_medicina, 2023, doi:10.3390/medicina59030536_

Round 1

Reviewer 1 Report

The authors try to use deep learning to predict the TERT promoter mutation based on 13 consecutive surgically resected thyroid cancers with TERT promoter mutations and 12 randomly selected surgically resected thyroid cancers. My major concerns are the following: First of all, deep learning to predict the gene mutation based on morphological data need much big number of cases and multi-module analysis, as well as wet lab verification on big number of cases. Secondly, I don’t see any advantage of this method in clinical application, given that the PCR can do much better job than this method once you have the pathological sample. Thirdly, the chosen of samples has only one TERT mutation difference, but the other gene status might be very different, which could also contribute to the morphology of cancer big time.

Minor points:

1.      The English need to be improved.

2.      The authors wrote: “During the practice, we retrospectively searched 80 consecutive surgically resected 67 thyroid cancer cases from 2016 to 2021 whose samples underwent TERT promoter PCR tests, and found 13 (16.3%) cases with TERT promoter mutations (either C228T or C250T). TERT promoter mutational statuses were confirmed by real-time polymerase chain reaction (PCR)”, what happened to the other 83.7% of samples, why not use those as TERT wild type control?

Author Response

Thank you for giving us the opportunity to revise our manuscript. The reviewers’ comments were very helpful. Our responses to the comments are as follows. All the changes in the manuscript  were highlighted in yellow.

Reviewer #1:

Major points

  1. First of all, deep learning to predict the gene mutation based on morphological data need much big number of cases and multi-module analysis, as well as wet lab verification on big number of cases.

Thank you for the comment. We agreed that training a deep learning model needs much large number of cases. However, TERT promoter mutation is not a frequent event in thyroid cancer. Thus, collecting a large number of thyroid cancer cases with TERT promoter mutation was difficult in this study.

Given our knowledge on issues regarding case number, we focused on smart-sized cases confirmed by PCR testing and an efficient learning approach by transfer learning. Furthermore, this work focused on tile-level rather than case-level prediction using deep learning. Thus, the number of tile images (Table 1) was sufficient to train the deep learning model. Although many relevant studies assign the same label to every patch in the tumor region of a WSI, this approach suffers from noisy training.

Therefore, to obtain good quality data from a limited number of TERT-positive cases, two pathologists who are experienced in thyroid pathology involved in our study conducted fine-grained TERT ROI annotation in tumor areas. Using a smart-sized and good data set, the deep learning approach was able to differentiate the morphologic features at the tile level with 0.99 sensitivity for TERT mutation positivity.

We have added it to the manuscript (line 361-371).

  1. I don’t see any advantage of this method in clinical application, given that the PCR can do much better job than this method once you have the pathological sample. 

Thank you for the comment. As pointed by the reviewer, Real-time PCR testing and next-generation sequencing are currently being used to confirm TERT promoter mutation status (line 314-316). However, testing all thyroid cancers for the TERT promoter mutation might not be cost effective considering the low incidence of TERT promoter mutation in thyroid cancer. Therefore, predicting TERT promoter mutations via histologic images using a deep learning approach can be a useful screening tool. We have added it to the manuscript (line 316-319).

  1. The chosen of samples has only one TERT mutation difference, but the other gene status might be very different, which could also contribute to the morphology of cancer big time.

Indeed, among the 13 thyroid cancer cases with TERT promoter mutations, five cases subsequently underwent targeted next-generation sequencing at the request of the clinician (data not included). Other than TERT promoter hotspot mutations, pathogenic mutations in NRAS, TP53, BRAF, RB1 deletion, and NCOA4-RET fusion were detected in these cases. However, only TERT promoter mutations were recurrently detected in these cases. We therefore suggested that TERT promoter mutations were most closely associated with the histologic findings observed in the present study.

We have added it to the manuscript (line 334-340).

Minor points

  1. The English need to be improved.

We have again received the English editing service and submitted the certificate. Thank you.

  1. The authors wrote: “During the practice, we retrospectively searched 80 consecutive surgically resected 67 thyroid cancer cases from 2016 to 2021 whose samples underwent TERT promoter PCR tests, and found 13 (16.3%) cases with TERT promoter mutations (either C228T or C250T). TERT promoter mutational statuses were confirmed by real-time polymerase chain reaction (PCR)”, what happened to the other 83.7% of samples, why not use those as TERT wild type control?

Thank you for the comment, we randomly selected the 12 cases of out of 54 TERT wild-type thyroid cancer cases (the other 83.7% of samples) for the control (line 78-79). Considering a class-imbalance problem in training deep learning, we finally selected the number of TERT negative cases which is similar to those of positive ones. We have added it to the manuscript (line 79-81). 

Reviewer 2 Report

Dear Author

The subject of your research is an interesting one. Detecting TERT mutation in head and neck malignancies through AI. 

The research aim was TERT promoter mutational status detection through Artificial Intelligence (AI) in thyroid cancer  based on the histologic image which is an interesting one.

It is relevant and interesting.

It improves thyroid cancer detection speed and accuracy. Based on their own writing "TERT promoter mutations can be screened for in a highly sensitive 26 manner using histologic image analysis based on deep learning. This approach will help to better 27 classify thyroid cancer patients according to the biological behavior of tumors".

The paper is well written,  just a small English edit is required.

The text is clear and easy to read.

The conclusions are consistent with the evidence and arguments presented.

They address the main question posed.

Author Response

Thank you very much.

Reviewer 3 Report

The authors designed a two-step cascaded architecture using CNN and CENN, in order to predict the mutational status in cancer, which will help to better classify thyroid cancer patients according to the biologic behavior of tumors. The cascade deep learning approach was well conducted and the result was in line with previous studies that TERT promoter mutations usually accompany morphological changes. While there are some questions should be mentioned as below:

1)The credibility of this work still has to be interpreted with caution given the small sample size, therefore, a subsequent study with a larger cohort might be needed?

2)Line 72-74: “TERT promoter PCR testing was mainly performed for......showing aggressive clinical behavior”: Honestly, the TERT promoter mutation displays an association with aggressive types of thyroid cancer, including a sharply increased tumor recurrence and patient mortality. While the PCR tests were mainly performed for advanced individuals, which could lead to the selection bias (e.g., low-risk pathological type with TERT positive). Could you discuss more of this question? Or make a supplementary explanation in the Limitation Part.

3)This study provided a detailed methodologic illustration expect for the pathotyping of tissue slides. However, TERT promoter mutation is a rare genetic event in PTMC for instance. Therefore, the deep learning prediction will be made more accurate if a subgroup analysis were performed.

4)Line 93-98: Since there exists the intratumoral heterogeneity in thyroid cancer, the expression status and abundance of TERT promoter mutation could be different in distinct tumor area, I suggest that the author could validate the target gene expression through IHC staining.

Author Response

Thank you for giving us the opportunity to revise our manuscript. The reviewers’ comments were very helpful. Our responses to the comments are as follows. All the changes in the manuscript  were highlighted in yellow.

Reviewer #3:

  1. The credibility of this work still has to be interpreted with caution given the small sample size, therefore, a subsequent study with a larger cohort might be needed?

Thank you for the comment. We agreed that training a deep learning model needs much large number of cases. However, TERT promoter mutation is not a frequent event in thyroid cancer. Thus, collecting a large number of thyroid cancer cases with TERT promoter mutation was difficult in this study.

Given our knowledge on issues regarding case number, we focused on smart-sized cases confirmed by PCR testing and an efficient learning approach by transfer learning. Furthermore, this work focused on tile-level rather than case-level prediction using deep learning. Thus, the number of tile images (Table 1) was sufficient to train the deep learning model. Although many relevant studies assign the same label to every patch in the tumor region of a WSI, this approach suffers from noisy training.

Therefore, to obtain good quality data from a limited number of TERT-positive cases, two pathologists who are experienced in thyroid pathology involved in our study conducted fine-grained TERT ROI annotation in tumor areas. Using a smart-sized and good data set, the deep learning approach was able to differentiate the morphologic features at the tile level with 0.99 sensitivity for TERT mutation positivity.

We have added it to the manuscript (line 361-371).

  1. Line 72-74: “TERT promoter PCR testing was mainly performed for......showing aggressive clinical behavior”: Honestly, the TERT promoter mutation displays an association with aggressive types of thyroid cancer, including a sharply increased tumor recurrence and patient mortality. While the PCR tests were mainly performed for advanced individuals, which could lead to the selection bias (e.g., low-risk pathological type with TERT positive). Could you discuss more of this question? Or make a supplementary explanation in the Limitation Part.

Thank for the comment. We mainly performed PCR testing for TERT promoter mutation in thyroid cancer cases presenting with large tumor sizes showing aggressive behavior, which might lead to selection bias. Subsequent studies that include a larger number of thyroid cancer cases with a wider morphological spectrum should be performed to further validate the findings presented in this study.

We have added it to the manuscript (line 341-345).

  1. This study provided a detailed methodologic illustration expect for the pathotyping of tissue slides. However, TERT promoter mutation is a rare genetic event in PTMC for instance. Therefore, the deep learning prediction will be made more accurate if a subgroup analysis were performed.

Thank you for the comment. If the number of thyroid cancer cases were sufficient for the subclassification into histologic subtypes, the deep learning prediction can be conducted on each subtype and analyze performance according to the subgroup. In future works, we absolutely do consider the subgroup analysis after we collect a large number of cases. We have added it to the manuscript (line 346-349).

  1. Line 93-98: Since there exists the intratumoral heterogeneity in thyroid cancer, the expression status and abundance of TERT promoter mutation could be different in distinct tumor area, I suggest that the author could validate the target gene expression through IHC staining.

Unfortunately, we do not have TERT promoter IHC in our pathology lab. However, targeted next-generation sequencing was performed in 5 out of 13 TERT promoter mutant thyroid cancer cases in out cohort. In NGS analysis, TERT promoter mutations in four of the five cases who underwent targeted next-generation sequencing were found to be clonal events after considering tumor cellularity and variant allele frequency of the TERT promoter mutation (data not included), with only one case having a TERT promoter mutation determined to be a subclonal event.

We have added it to the manuscript (line 350-355).

Round 2

Reviewer 1 Report

In the revised manuscript, the authors have not addressed the questions.  The case number is still too low to train a deep learning system. The other mutated genes also contribute to the morphology, which disturb the deep learning system big time. For these reasons, it is not possible for this kind of deep learning to be applicable in clinic to replace the PCR, plus PCR is not expensive at all. I still don't see the clinical significance of this work.

Reviewer 3 Report

This manuscript has been more or less amended. I think this research is slightly applealing. However, the scientific value still remains to be improved.